# Metagenome fragment classification based on multiple motif-occurrence profiles

Naoki Matsushita, Shigeto Seno, Yoichi Takenaka and Hideo Matsuda

Department of Bioinformatic Engineering, Graduate School of Information Science and Technology, Osaka University, Yamadaoka, Suita, Osaka, Japan

## ABSTRACT

A vast amount of metagenomic data has been obtained by extracting multiple genomes simultaneously from microbial communities, including genomes from uncultivable microbes. By analyzing these metagenomic data, novel microbes are discovered and new microbial functions are elucidated. The first step in analyzing these data is sequenced-read classification into reference genomes from which each read can be derived. The Naïve Bayes Classifier is a method for this classification. To identify the derivation of the reads, this method calculates a score based on the occurrence of a DNA sequence motif in each reference genome. However, large differences in the sizes of the reference genomes can bias the scoring of the reads. This bias might cause erroneous classification and decrease the classification accuracy. To address this issue, we have updated the Naïve Bayes Classifier method using multiple sets of occurrence profiles for each reference genome by normalizing the genome sizes, dividing each genome sequence into a set of subsequences of similar length and generating profiles for each subsequence. This multiple profile strategy improves the accuracy of the results generated by the Naïve Bayes Classifier method for simulated and Sargasso Sea datasets.

## INTRODUCTION

Complete genome sequences have been obtained from thousands of organisms that are generally well-studied and cultivable. Recently, however, an increasing number of studies have performed metagenomic analysis focused on uncultivable microbes (*Tringe & Rubin, 2005*). To analyze metagenomic data, 16S rRNA sequences have been widely used for phylogenetic systematics (*Wang & Qian, 2009*). However, 16S rRNA is not directly involved in the biological functions of microbes. Thus, 16S rRNA can be used to classify microbial communities (*Koslicki, Foucart & Rosen, 2014*), but it cannot be used to identify novel biofunctional genes.

As an alternative, metagenomic analysis targeting DNA sequenced-reads extracted directly from a microbial community has been developed (*Robe et al., 2003*). For the following discussion, the term metagenomic data refers to a set of sequenced-reads. Metagenomic analysis is the analysis of environmental microbial flora, referred to

Corresponding author
Hideo Matsuda,
matsuda@ist.osaka-u.ac.jp

as the metagenome. Using this method, many microbes can be directly sequenced simultaneously, and their sequenced-read data are analyzed. The first step in metagenomic analysis is the classification of the data into available complete genomes (reference genomes). Sequenced-reads obtained from particular environments are classified to identify their derivation. High classification accuracy is required for the further analyses that are conducted after classification. There are two types of classification methods: composition-based methods, such as the Naïve Bayes Classifier (NBC) (*Rosen et al., 2008*), RAIphy (*Nalbantoglu et al., 2011*), WGSQuikr (*Koslicki, Foucart & Rosen, 2014*) and GSTaxClassifier (*Yu et al., 2009*), and comparison-based methods, such as MEGAN (*Huson et al., 2007*), BLAST (*Altschul et al., 1990*), Treephyler (*Schreiber et al., 2010*), SPANNER (*Porter & Beiko, 2013*) and MyTaxa (*Luo, Rodriguez-R & Konstantinidis, 2014*).

A comparison-based method scans the reference genomes registered in a database and identifies the genomes that are similar to the queried read. A composition-based method extracts sequence characteristics from both the sequenced-reads and reference genomes. Composition-based methods generally classify sequenced-reads more rapidly than comparison-based methods (*Rosen et al., 2008*). Moreover, a comparative study of these methods showed that NBC outperformed all other composition-based methods in terms of combined sensitivity and precision (*Bazinet & Cummings, 2012*).

Composition-based methods use sequence characteristics consisting of *motifs*—sequence patterns of short and fixed-length and *profiles*—the occurrence frequencies for each motif in each reference genome. NBC typically uses motifs longer than 12-mer, whereas WGSQuikr, which is based on a *k*-mer matrix, uses motifs shorter than 7-mer (*Koslicki, Foucart & Rosen, 2014*). Then, NBC calculates scores according to the likelihood that a sequenced-read is derived from a particular reference genome. Each read is assigned to the genome displaying the highest score.

NBC assumes that the occurrence probability of each motif is independent. This assumption may appear unrealistic, but it has been reported that NBC generally exhibits a high classification accuracy (*Rish, 2001*). However, larger genomes have more motifs than smaller genomes because the motifs have been extracted from their sequences. Thus it is more likely to match motifs with larger genomes in a sequenced-read during metagenome classification. The sizes of microbial genomes vary widely. For this reason, bias may be introduced that results in erroneous classifications (Fig. 1).

First, consider the two very similar motif-occurrence profiles for genomes "A" and "B" shown in Fig. 1A. Motif-occurrence profile "A" is composed of 1 red, 2 blue, 1 green, and 1 yellow, whereas motif-occurrence profile "B" is composed of 1 red, 2 blue, 1 green, and 2 yellow. In this situation, a sequenced-read (DNA fragment) composed of 1 red, 1 blue, and 1 green is assigned to genome "A". In this example, the read is assigned to the smaller genome.

Alternatively, suppose that two motif-occurrence profiles, labeled as genomes "C" and "D" in Fig. 1B, display very similar abundance ratios for the motifs. Motif occurrence profile "C" is composed of 5 red, 4 blue, 3 green, and 6 yellow, whereas motif-occurrence profile "D" is composed of 1 red, 2 blue, 1 green, and 2 yellow. In this situation, a

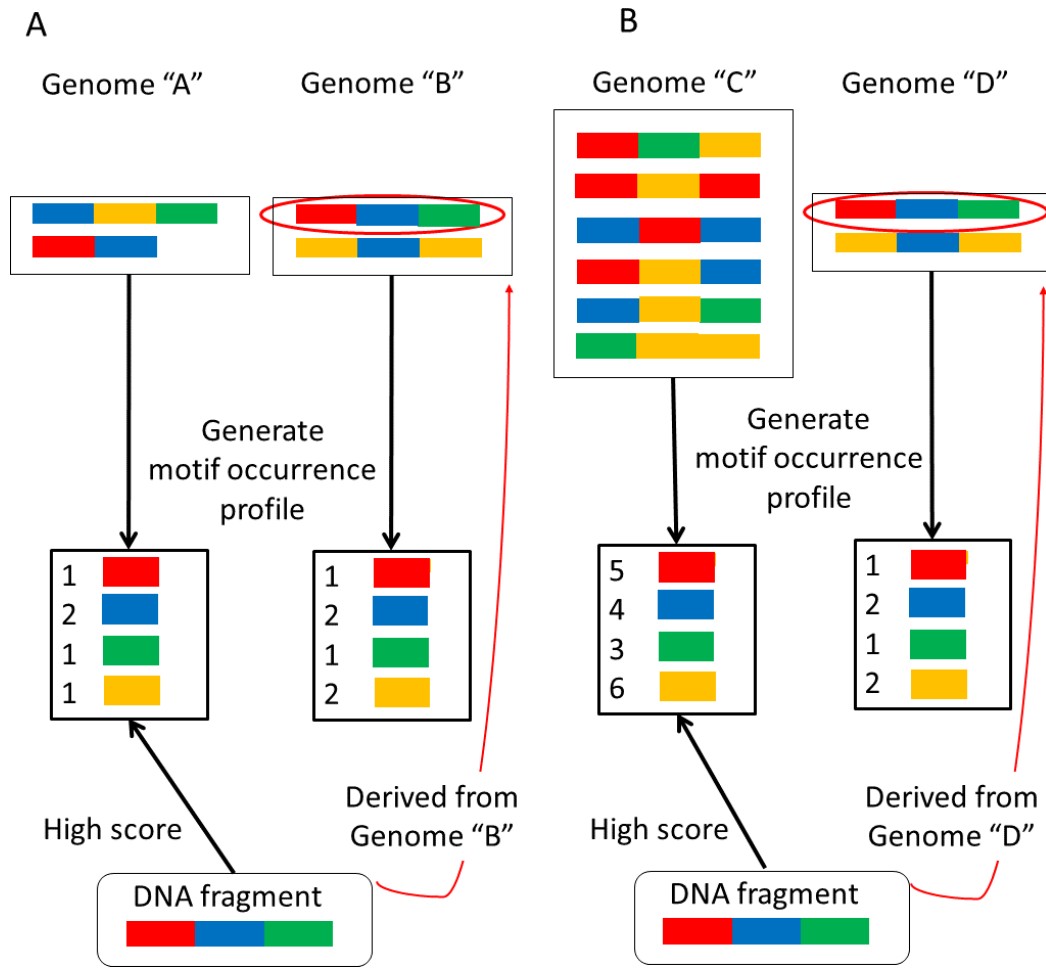

**Figure 1 Examples of erroneous classifications.** (A) A sequence-read (DNA fragment) derived from the larger genome "B" is assigned to the smaller genome "A." (B) A sequence-read derived from the smaller genome "D" is assigned to the larger genome "C".

sequenced-read composed of 1 red, 1 blue, and 1 green is assigned to genome "C", the larger genome.

It has been reported that NBC incorrectly assigns some reads that are derived from smaller genomes to larger genomes (*Rosen et al., 2008*). In the present report, we propose a more accurate classification method. The first step in analyzing metagenomic data is sequenced-read classification into reference genomes from which each read can be derived. A high classification accuracy is necessary for subsequent metagenomic analyses. In the following section, we describe an updated NBC method to reduce the erroneous classification depicted in Fig. 1 by reducing the differences in the reference genome sizes. Moreover, we validated our method by demonstrating an improvement in the classification performance using simulated and real datasets.

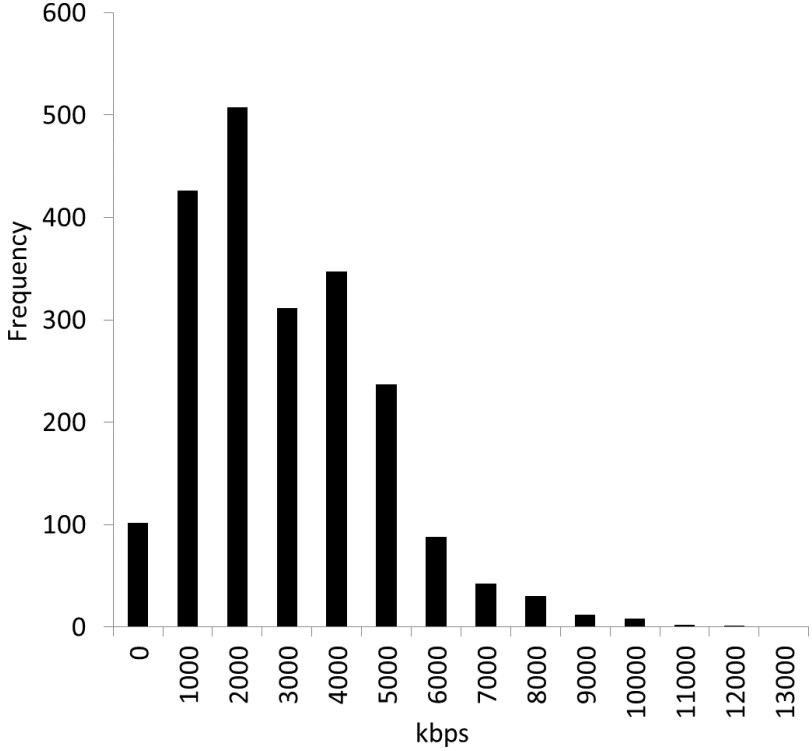

**Figure 2 Distribution of 2,112 bacterial genome sizes.** The average genome size was 3,451 kbps, and the standard deviation was 1,882 kbps. The range of the size difference was approximately 93-fold.

## MATERIALS AND METHODS

We obtained reference genome sequence data file all.fna.tar.gz in December 2012 from the NCBI Microbial Genomes FTP site: ftp://ftp.ncbi.nih.gov/genomes/Bacteria/. At that time, the number of genome sequences was 2,112 (see the list of strains in the Data S1). Figure 2 shows the distribution of the genome sizes of these 2,112 bacterial genomes, demonstrating large differences in their sizes. The average genome size was 3,451 kbps, and the standard deviation was 1,882 kbps. The largest genome was *Sorangium cellulosum* So ce 56, and the smallest genome was *Candidatus Tremblaya princeps* PCIT. The range of the size difference was approximately 93-fold.

We have updated the NBC method (*Rosen et al., 2008*) to reduce the erroneous classification illustrated in Fig. 1. NBC is a statistical method based on Bayes' theorem. Bayes' theorem involves the updating of a priori probabilities after obtaining new information. Bayes' theorem is expressed as shown in Eq. (1).

$$P(A|B) = \frac{P(A)P(B|A)}{P(B)}.$$

(1)

NBC applies Bayes' theorem to metagenome fragment classification. It defines $C_1, C_2, C_3, \ldots, C_P$ as $P$ genome classes and $\mathbf{d} = \{d_1, d_2, d_3 \ldots d_K\}$ as the feature vector

composed of a set of motifs (fixed-length sub-sequences patterns) of a DNA fragment (i.e., sequenced-read):

$$P(C_i|d) = \frac{P(C_i)P(d|C_i)}{P(d)}. \tag{2}$$

Equation (2) shows the posterior probability of a particular class $C_i$. To simplify Eq. (2), NBC assumes that the probability of each motif's appearance is independent, resulting in the following variation shown in Eq. (3):

$$P(C_i|d) = \frac{P(C_i)\prod_{j=1}^{K} P(d_j|C_i)}{P(d)}. \tag{3}$$

In Eq. (3), NBC also assumes that the occurrence probability of a feature vector expressed by $P(\mathbf{d})$, is a constant. In addition, the targets of our analysis are not metagenomic datasets generated from specific environments, but, rather, metagenomic datasets generated from various environments. Therefore, we assume that the occurrence probabilities in genomes $P(Ci)$ are equal because the actual composition of the metagenome is unknown. An equation that maximizes Eq. (3) is provided in Eq. (4), which shows the $i$th class from which a DNA fragment may be derived:

$$\hat{C} = \arg\max_i \prod_{j=1}^{K} P(d_j|C_i). \tag{4}$$

To avoid underflow, the equation is computed as the logarithm

$$\hat{C} = \arg\max_i \sum_{j=1}^{K} \log P(d_j|C_i). \tag{5}$$

The classification method consists of four steps:

    Step 1. Generate motif-occurrence profiles from each genome sequences.
    Step 2. Extract feature vectors from each DNA fragment.
    Step 3. Calculate the scores of each genome for every DNA fragment in the metagenome.
    Step 4. Assign each fragment to the genome with the highest score.

In Step 4, NBC assigns every fragment to the genome with the highest score without a threshold. We have updated the NBC method using multiple profiles for each reference genome, referred to as *NBC-MP* (multiple profiles). NBC generates a single motif-occurrence profile, the recorded frequencies of the fixed-length sub-sequences of a reference genome, for each genome. In NBC-MP, each reference genome sequence is separated into multiple sub-sequences of similar length according to the size of the given genome. Thereafter, NBC-MP generates motif-occurrence profiles for each sub-sequence. Thus, the NBC-MP method includes multiple profiles from each genome (Fig. 3).

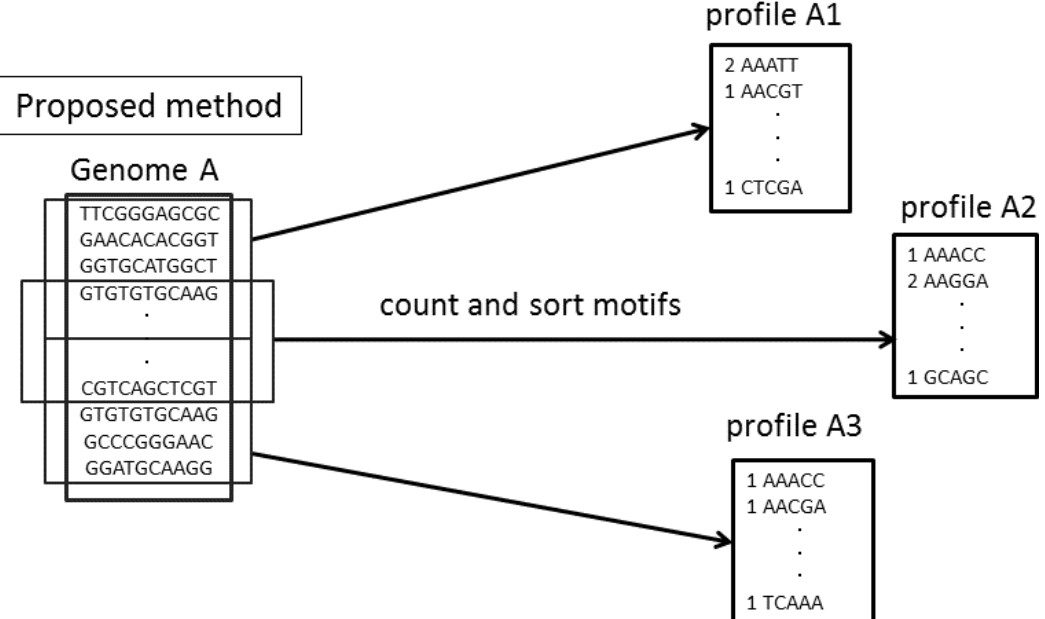

**Figure 3 Differences in the motif profiles between the NBC (previous) and NBC-MP (proposed) methods.** NBC-MP (proposed) generates multiple profiles from each genome, whereas NBC (previous) generates a single profile for each genome.

NBC-MP consists of the following six steps:

Step 1. Separate the genomes into groups according to their genome size.

Step 2. Except for Group 1, which contains the smallest genomes, divide the genomes into sub-sequences.

Step 3. Generate motif-occurrence profiles from each sub-sequence.

Step 4. Extract feature vectors from each DNA fragment.

Step 5. Calculate the scores of each sub-sequence for every fragment in the metagenome.

**Table 1 Groups according to their genome size.** In the NBC-MP method, 2,112 bacterial genomes were classified into four groups according to their size.

|  | Group 1 | Group 2 | Group 3 | Group 4 |
|---|---|---|---|---|
| # genomes | 527 | 818 | 584 | 183 |
| # partitions | 1(no division) | 3 | 5 | 7 |
| # bases before division | 140 k–2,000 k | 2,001 k–4,000 k | 4,001 k–6,000 k | 6,001 k–12,912 k |
| # bases after division | 140 k–2,000 k | 1,000 k–2,000 k | 1,333 k–2,000 k | 1,500 k–3,228 k |

Step 6. Assign each fragment to the sub-sequence with the highest score and combine the assigned fragments according to each original genome.

In Step 5, similar to the NBC method, the NBC-MP method calculates scores for each sub-sequence individually. In Step 6, NBC-MP assigns every fragment to the sub-sequence displaying the highest score without a threshold, just as in Step 4 of NBC. After the assignment step, NBC-MP combines the assigned fragments according to each original genome by compiling the separated sub-sequences into their original genome.

In detail, in Groups 2 through 4 in Table 1, assuming that the partitions (the number of sub-sequences for each reference genome) are $n$ and the length of the reference genome is $L$, the size of each sub-sequence is $2*L/(n + 1)$. Therefore, the adjacent sub-sequences contain overlapping regions of length $L/(n + 1)$. For example, given a 3 Mb genome belonging to Group 2 (the number of the partitions is 3), our method divides this genome into 3 sub-sequences (the size of each sub-sequence is 1.5 Mb, and the adjacent sub-sequences contain 750 kb overlapping regions).

We classified the 2,112 bacterial reference genomes into four groups according to their size (Table 1). We did not separate the genome sequences belonging to Group 1 because this group contained the smallest genome sequences. Then, we separated the genome sequences belonging to Group 2 into three sub-sequences, those in Group 3 into five sub-sequences, and those in Group 4 into seven sub-sequences. As shown in Fig. 2, the original genome sizes largely vary. The ratio of the largest genome to the smallest genome was 93. The larger genomes needed to be separated into more sub-sequences to reduce the genome size difference. However, excessive separation would produce many sub-sequences and require an enormous amount of computational time. Table 1 summarizes the four groups and the partitions used in the proposed method according to their genome size. We empirically selected these parameters. Separating the genome sequences reduced the genome size ratio to approximately 23.

## RESULTS AND DISCUSSION

### Experiment using a simulated dataset

We generated simulated datasets of sequenced-reads using wgsim (available at: https:// github.com/lh3/wgsim) from the Samtools package (*Li et al., 2009*) using the option "-N 100 -l <length> -e 0 -r <rate> -R 0.0" (other options were set at the default values). We changed <length> to 25, 100, or 500 (read length) and <rate> to 0.0, 0.01, or 0.02

**Peer**J

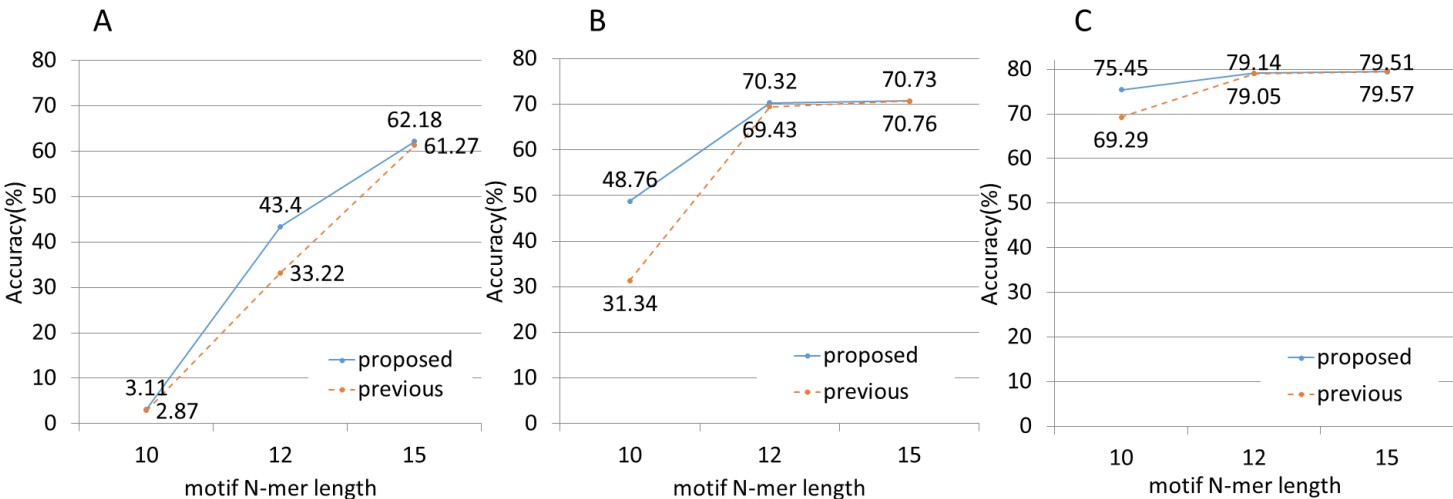

**Figure 4 Classification accuracy performance.** (A) The fragment length was 25. The NBC-MP method (proposed) more accurately classified the fragments in this case. (B) The fragment length was 100. The NBC-MP method (proposed) more accurately classified the 10-mer and 12-mer motifs. (C) The fragment length was 500. The NBC-MP method (proposed) more accurately classified the 10- and 12-mer motifs. However, the NBC (previous) method classified the 15-mer motifs slightly more accurately.

(sequence error rate). For each option, we used the input file for each reference genome sequence, repeatedly executed these inputs for the 2,112 genome sequences, and merged all of the output reads into a single data file, which we used as the simulated dataset. We selected three motifs lengths 10-, 12-, and 15-mer just as in the NBC method (*Rosen et al., 2008*), to compare the classification results between NBC and NBC-MP.

First, we applied the NBC and NBC-MP methods to three sets of simulated data containing no sequenced-read errors. We defined accuracy as shown in Eq. (6). These experiments revealed the performances of the methods with respect to classification accuracy, as shown in Fig. 4. The NBC-MP method outperformed the NBC method, except for the condition of 500 bp fragments and 15-mer motifs.

$$\text{Accuracy} = \frac{\text{number of DNA fragments assigned correctly}}{\text{number of total DNA fragments}}. \tag{6}$$

Next, we applied both methods to simulated data containing sequenced-read errors (Fig. 5). These errors included only base substitutions and did not include insertions or deletions. The NBC-MP method more accurately classified the fragments compared to the NBC method in the presence of increasing sequence error rates.

### Real dataset from a Sargasso Sea sample

We conducted experiments using the Sargasso Sea dataset produced by Sanger sequencing (*Venter et al., 2004*). These sequence data were retrieved from DDBJ/EMBL/GenBank (accession number AACY01000000, Locus CH004737 to CH236877). We selected the first 10,000 fragments for analysis as used in the MEGAN and NBC analyses (*Huson et al., 2007*; *Rosen et al., 2008*). The average fragment lengths was 889 bp. Figure 6 shows the distribution of the fragment lengths.

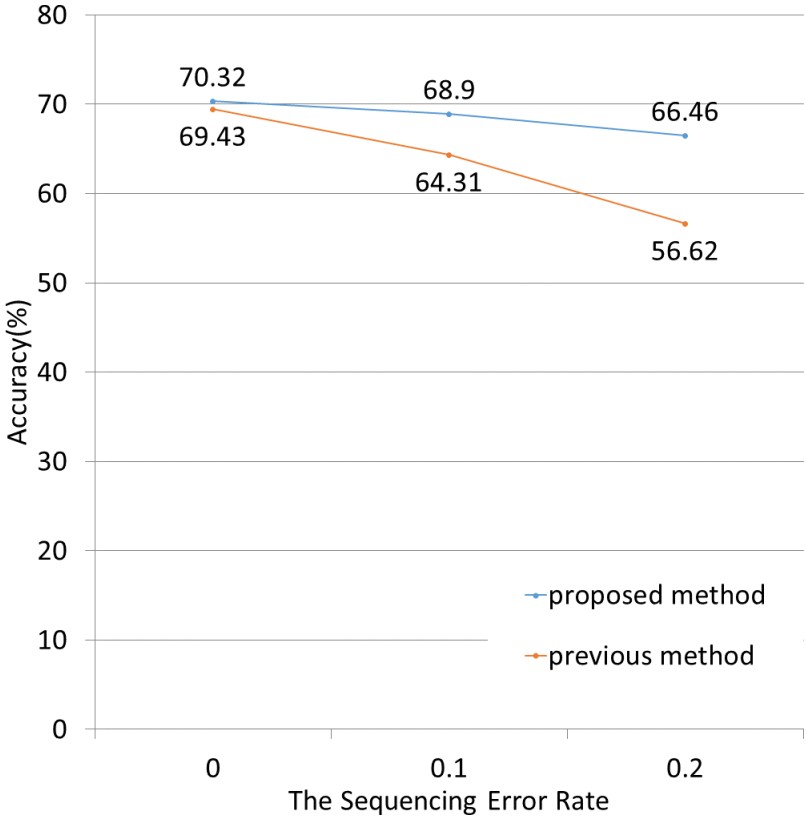

**Figure 5 Classification accuracy for different sequence error rates.** The lengths of the motifs and fragments were fixed at 12 and 100, respectively. The sequence errors rate was set at 0%, 1%, or 2%. The NBC-MP method (proposed) outperformed the NBC method (previous) with respect to accuracy. Moreover, the improvement using the NBC-MP method was greater in the presence of more frequent sequence errors.

Table 2 shows the results of the NBC and NBC-MP classifications of 15-mer motifs in the dataset. As shown in Table 2A, the NBC method tended to give preference to larger genomes; thus, the number of fragments classified into each genome was greater in the larger genomes (e.g., *Trichodesmium erythraeum* IMS101, *Clostridium beijerinckii* NCIMB 8052, *Flavobacterium johnsoniae* UW101, and *Flexibacter litoralis* DSM 6794). In contrast, using the NBC-MP method, the fragments tended to be classified into smaller genomes (e.g., *Arcobacter* L, *alpha proteobacterium* HIMB5, and *Candidatus pelagibacter ubique* HTCC1062). To evaluate the accuracies of these two methods, we referred to the results of another comparison-based tool, MEGAN. MEGAN compares the set of fragments against databases of known sequences using BLAST in a preprocessing step and assigns each fragment to the lowest common ancestor (LCA) (*Huson et al., 2007*). In this study, we performed a strain-level assignment of the fragments. Thus, we adopted BLAST, the first step of MEGAN, because it can classify metagenomic fragments without being influenced by the differences in the genome sizes.

We generated a bacterial BLAST database using makeblastdb (available at: ftp://ftp.ncbi. nlm.nih.gov/blast/executables/blast+/LATEST/) from the blast+ package using the option
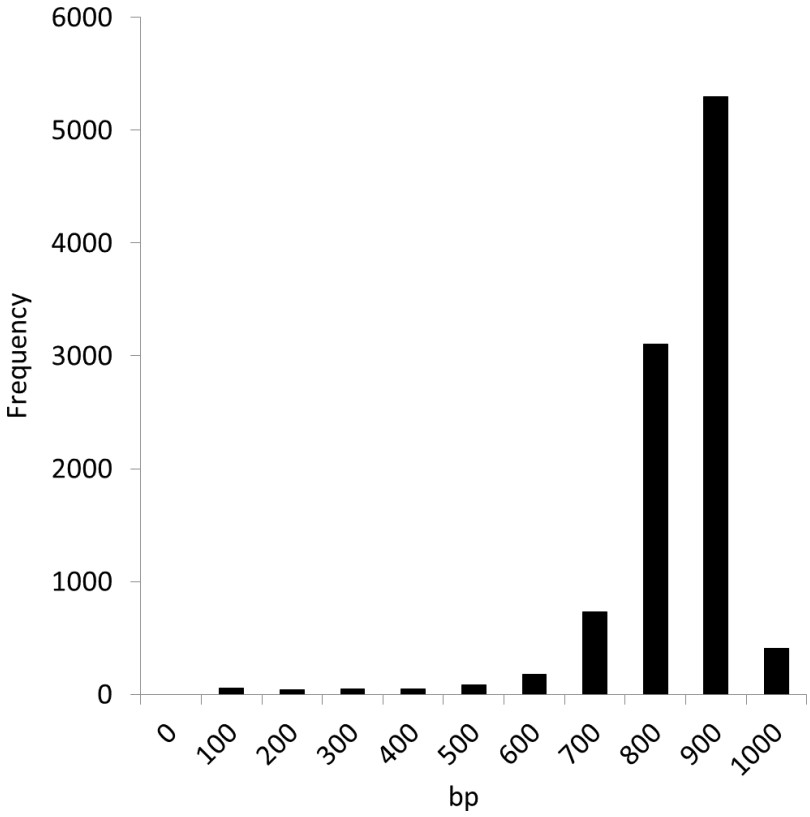

**Figure 6 Length distribution of the first 10,000 fragments of the Sargasso Sea dataset.**

"-in <input file> -dbtype nucl -hash_index -max_file_sz '10GB'" (the other options were set at the default values). We integrated the 2,112 bacterial reference genomes into a single multifasta format file and used it as the input file. After generating this BLAST database, we ran blastn (available at: ftp://ftp.ncbi.nlm.nih.gov/blast/executables/blast+/LATEST/) from the blast+ package using the option "-query <input file> -db <BLAST database> -outfmt 5 -out <output file>" (the other options were set at the default values). We used the top 10,000 fragments of the Sargasso Sea dataset as the input file. The BLAST search detected *Burkholderia* 383, *Shewanella* ANA 3, and *Shewanella oneidensis* MR 1 as the three most abundant strains in the dataset (see the list of strains detected by the BLAST search in the Data S2).

After running this BLAST search, we calculated Spearman's correlation coefficient against the NBC and NBC-MP methods within 12 common strains in the top 30 strains of the BLAST, NBC, and NBC-MP classifications. The Spearman's correlation coefficient between the BLAST and NBC classifications was 0.825, whereas the coefficient between the BLAST and NBC-MP classifications was 0.944. The BLAST result corresponded well to the results of the NBC-MP method compared with the NBC method (Table 2B). Although the actual composition of the metagenome was unknown in the dataset, we conclude that the NBC-MP classification result is more suitable than the NBC classification result because the different method independently agreed with the NBC-MP classification result.

**Matsushita et al. (2014), *PeerJ*, DOI 10.7717/peerj.559**  10/13

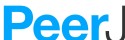

**Table 2 Classification results in the first 10,000 fragments in the Sargasso Sea dataset.** (A) The abundance of the strains detected using the NBC, NBC-MP, and BLAST methods were ranked in decreasing order of their genome size for the eight most abundant strains detected using the BLAST method. (B) The detected strains were ranked in order of abundance according to the NBC (left) and NBC-MP (right) methods for the eight most abundant strains detected using the BLAST method.

**(A)**

| | | # fragments | | | |
| Strain | Genome size | NBC | NBC-MP | BLAST | BLAST rank |
|---|---|---|---|---|---|
| *Burkholderia* 383 | 8.7 | 1,901 | 1,856 | 1,831 | 1 |
| *Burkholderia cenocepacia* J2315 | 7.8 | 172 | 176 | 148 | 8 |
| *Trichodesmium erythraeum* IMS101 | 7.5 | 336 | 5 | 1 | – |
| *Clostridium beijerinckii* NCIMB 8052 | 5.9 | 1,026 | 70 | 2 | – |
| *Flavobacterium johnsoniae* UW101 | 5.9 | 246 | 4 | 4 | – |
| *Shewanella* ANA 3 | 5.1 | 986 | 981 | 963 | 2 |
| *Shewanella oneidensis* MR 1 | 5 | 292 | 300 | 312 | 3 |
| *Flexibacter litoralis* DSM 6794 | 4.8 | 286 | 25 | 3 | – |
| *Shewanella* MR 7 | 4.7 | 203 | 202 | 231 | 5 |
| *Shewanella* MR 4 | 4.6 | 241 | 248 | 206 | 6 |
| *Clostridium botulinum* B Eklund 17B | 3.8 | 105 | 179 | 1 | – |
| *Arcobacter* L | 2.9 | 156 | 311 | 2 | – |
| *alpha proteobacterium* HIMB5 | 1.4 | 99 | 518 | 237 | 4 |
| *Candidatus pelagibacter ubique* HTCC1062 | 1.3 | 56 | 281 | 189 | 7 |

**(B)**

| NBC method | | | NBC-MP method | | |
| Strain | # fragments | BLAST rank | Strain | # fragments | BLAST rank |
|---|---|---|---|---|---|
| *Burkholderia* 383 | 1,901 | 1 | *Burkholderia* 383 | 1,856 | 1 |
| *Clostridium beijerinckii* NCIMB 8052 | 1,026 | – | *Shewanella* ANA 3 | 981 | 2 |
| *Shewanella* ANA 3 | 986 | 2 | *alpha proteobacterium* HIMB5 | 518 | 4 |
| *Trichodesmium erythraeum* IMS101 | 336 | – | *Arcobacter* L | 311 | – |
| *Shewanella oneidensis* MR 1 | 292 | 3 | *Shewanella oneidensis* MR 1 | 300 | 3 |
| *Flexibacter litoralis* DSM 6794 | 286 | – | *Candidatus pelagibacter ubique* HTCC1062 | 281 | 7 |
| *Flavobacterium johnsoniae* UW101 | 246 | – | *Shewanella* MR 4 | 248 | 6 |
| *Shewanella* MR 4 | 241 | 6 | *Shewanella* MR 7 | 202 | 5 |
| *Shewanella* MR 7 | 203 | 5 | *Clostridium botulinum* B Eklund 17B | 179 | – |
| *Burkholderia cenocepacia* J2315 | 172 | 8 | *Burkholderia cenocepacia* J2315 | 176 | 8 |
| *Arcobacter* L | 156 | – | *Clostridium beijerinckii* NCIMB 8052 | 70 | – |
| *Clostridium botulinum* B Eklund 17B | 105 | – | *Flexibacter litoralis* DSM 6794 | 25 | – |
| *alpha proteobacterium* HIMB5 | 99 | 4 | *Trichodesmium erythraeum* IMS101 | 5 | – |
| *Candidatus pelagibacter ubique* HTCC1062 | 56 | 7 | *Flavobacterium johnsoniae* UW101 | 4 | – |

## CONCLUSION

In this report, we have addressed a difficulty with metagenome analysis caused by large differences in genome size. We have proposed a modified method, termed NBC-MP (multiple profiles), based on the Naïve Bayes Classifier (NBC) method. NBC-MP improves the accuracy of the NBC method for both simulated and real datasets. Furthermore, NBC-MP successfully classified DNA fragments containing sequenced-read errors. NBC-MP requires additional computational time due to the separation of the genomes. The increase in calculation time is approximately 4-fold compared with the original NBC method. However, the computation of the score for each fragment is independent. For this reason, we could parallelize the scoring method, shortening the running time according to the computational resources. Additionally, the number of groups and partitions of genomes for multiple profiles must be determined in advance. These numbers were empirically determined in this study and may affect the accuracy of the classification. Future studies will be conducted to reduce the computational time and to optimize the numbers of groups and genome partitions for maximal accuracy.

### Funding

This research used computational resources of the K computer and other computers of the HPCI system provided by the AICS and University of Tokyo through the HPCI System Research Project (Project ID: hp140230 and hp140118), and is supported in part by JSPS KAKENHI 26280106. The funders had no role in study design, data collection and analysis, decision to publish, or preparation of the manuscript.

### Grant Disclosures

The following grant information was disclosed by the authors:
HPCI System Research Project: hp140230, hp140118.
JSPS KAKENHI: 26280106.

### Competing Interests

The authors declare there are no competing interests.

### Author Contributions

- Naoki Matsushita conceived and designed the experiments, performed the experiments, analyzed the data, contributed reagents/materials/analysis tools, wrote the paper, prepared figures and/or tables, reviewed drafts of the paper.
- Shigeto Seno and Yoichi Takenaka conceived and designed the experiments, contributed reagents/materials/analysis tools, wrote the paper, reviewed drafts of the paper.
- Hideo Matsuda conceived and designed the experiments, contributed reagents/materials/analysis tools, wrote the paper, prepared figures and/or tables, reviewed drafts of the paper.

## Supplemental Information

Supplemental information for this article can be found online at http://dx.doi.org/10.7717/peerj.559#supplemental-information.

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
