# Peer review of "Metagenome fragment classification based on multiple motif-occurrence profiles"

_PeerJ, doi:10.7717/peerj.559_

## Round 0.1 · original submission · Major Revisions

· Academic Editor

Major Revisions

As you will find from the comments below, all of the three reviewers admit the value of your approach but two of them give some criticism on your work. Please read carefully the reviewers' comments and revise the manuscript accordingly. For my own comment, more explanation will help readers to understand how the performance was evaluated. Perhaps, more detailed comparison with MEGAN's is necessary, too.

Reviewer 1 ·

Basic reporting

No Comments.

Experimental design

No Comments

Validity of the findings

No Comments.

Additional comments

This paper presents a method for improving the naive Bayse-based method by considering the original bacterial genome size for metagenome fragment classification.
The authors showed that their new method is very simple but effective through experiments.

The manuscript is easy to read, and has no errors.

It would be more interesting if the authors add more discussions on the relationship between the genome size and P(C_i), as it will justify their method more theoretically.

Reviewer 2 ·

Basic reporting

This paper describes a new method for classifying sequence reads from metagenome samples. It improved classification performance compared with the previous Naïve Bayes Classifier by using multiple sets of occurrence profiles based on the genome size. The authors tackle an important problem of the metagenome analysis and the proposed method is potentially useful. However, several points need to be addressed.

First, the manuscript would benefit from some copy editing to correct English phrasing. For example, ‘because’ in the 2nd line of page 5 may not be necessary. Such English editing is necessary in the whole manuscript.

The manuscript also needs more description regarding introduction and methods. For example, in the introduction, the authors reasonably describe a background of this paper, but do not mention the aim, scope and achievement of this paper. It would be better to add such information at the end of the introduction.

In the methods or results section, it would be better to add some description about how the score is used to assign species. Is it just a top hit? And if so, are there any thresholds to assign or not assign the genome?

In COCLUSION, the authors discuss about the increase of computation time compared with the original NBC. I wonder the proposed method can benefit from parallel computing by dividing occurrence profiles and think the authors can add some discussion about it rather than just mentioning 4-fold increase of computation time.

Experimental design

Among several parameters including the number of groups, partitions and based length in the partitions, the authors evaluated only motif N-mer length in 10, 12, 15. Although the authors leave their evaluation as a future work, it would be nice to add some basic ideas and background of using the current parameters.

As a real application, the authors applied the proposed method to the NGS data from Sargasso Sea samples. However, there are no description about parameters used in this application. In addition, they compared their results with those by NBC based on MEGAN results and concluded that NBC-MP corresponds more to MEGAN than NBC. The author mention only top 3 hits, but more comprehensive comparison will be possible by, for example, Spearman’s correlation coefficient. It would be nice if the author could add such a comparison.

Validity of the findings

No comments.

Additional comments

No comments.

Reviewer 3 ·

Basic reporting

In this paper, the authors developed a new algorithm for meatgenomic reads classifier to identify their derivation from reference microbial genome sequences. This algorithm, the Naïve Bayes Classifier Multiple Profiles (NBC-MP), was modified from the previously developed Naïve Bayes Classifier (NBC), and in consequence, the accuracy of reads classification becomes improved not only for artificially generated sequence data but also for real metagenomic data. This reviewer fully agrees with the importance of the classification algorithm for metagenomics studies.

Experimental design

1. There is no precise description about the preparation of multiple profiles. The authors should be described more details how to divide the reference genome sequence? For instance, is there overlap region between adjacent groups? How was the overlap length?
2. In this paper, reference genome sequence was divided into two or more groups if the genome sequence was longer than 2Mb. However, how to determine or optimize the divided length? When the divided length is changed to more short or long, how about the accuracy of classification, increased or decreased? More discreet discussion should be needed.
3. If the genome length is longer than 2Mb (Group2~4), how the scores are calculated? Summation of all scores calculated from each group? The authors should be described more details how to calculate the score.

Validity of the findings

No Comments.

Additional comments

The following minor changes are needed for publication.
1. The authors should change the sub-title of the second chapter of Results and Discussions “Real NGS data from Sargasso Sea”, because the Sargasso Sea metagenomic study was done by Sanger type sequencer not by NGS.
2. Please denote the more information about the reference genome data of microbes. The authors described only “NCBI” and “December 2012”. Where did these data come from? NCBI GenBnak or RefSeq? What was the database version?
3. In table 1, base after division of Group3 and Group4, a comma should be inserted into a numeral both 2000K and 3228K (decimal separator).

---

## Round 0.2 · Minor Revisions

· Academic Editor

Minor Revisions

One of the reviewers point out a few minor points. Please confirm if you agree to the reviewer's opinion. If you agree, please submit the re-revised manuscript.

Reviewer 1 ·

Basic reporting

No comments.

Experimental design

No comments.

Validity of the findings

No comments.

Additional comments

No comments.

Reviewer 2 ·

Basic reporting

The authors reflected all the comments I have made.

Experimental design

No Comments.

Validity of the findings

No Comments.

Additional comments

L92: the targets of our analysis targets -> the targets of our analysis?

L128: It would be better to add an explanation of the number of groups. Probably just adding a reference to Table 1 would be enough.

L129: What is the difference between the partition and the number of sub-sequences? I assume that they are same, but if the number of sub-sequences is (n+1)/2 as indicated, then it will be 2 not 3 when the partition is 3.

Fig. 3 caption: right -> bottom, left -> top?
Table 2 caption: the 10 most abundant -> the eight most abundant?

Reviewer 3 ·

Basic reporting

In a newly revised manuscript, Matsushita et al. describe their newly developed algorithm for metgenomic reads classification.
The authors have responded well to my previous concerns.

I recommend the manuscript for publication.

Experimental design

No comments.

Validity of the findings

No comments.

---

## Round 0.3 · accepted · Accept

· Academic Editor

Accept

I confirm that you have addressed all the points raised by the reviewer properly and that the re-revised manuscript is now acceptable. Congratulations!